# A Meta-Analysis of the Impact of Nutritional Supplementation on Osteoarthritis Symptoms

**DOI:** 10.3390/nu14081607

**Published:** 2022-04-12

**Authors:** Sylvain Mathieu, Martin Soubrier, Cedric Peirs, Laurent-Emmanuel Monfoulet, Yves Boirie, Anne Tournadre

**Affiliations:** 1Service de Rhumatologie, Centre Hospitalier Universitaire Gabriel Montpied, F-63000 Clermont-Ferrand, France; msoubrier@chu-clermontferrand.fr (M.S.); atournadre@chu-clermontferrand.fr (A.T.); 2Neuro-Dol, Inserm, CHU Clermont-Ferrand, Université Clermont Auvergne, F-63000 Clermont-Ferrand, France; cedric.peirs@inserm.fr; 3CRNH Auvergne, Unité de Nutrition Humaine, INRAE, Université Clermont Auvergne, F-63000 Clermont-Ferrand, France; laurent-emmanuel.monfoulet@inrae.fr (L.-E.M.); yboirie@chu-clermontferrand.fr (Y.B.); 4Service de Nutrition Clinique, CHU Gabriel Montpied, F-63000 Clermont-Ferrand, France

**Keywords:** nutrients, osteoarthritis, efficacy, meta-analysis, systematic literature review

## Abstract

Conflicting evidence exists concerning the effects of nutrient intake in osteoarthritis (OA). A systematic literature review and meta-analysis were performed using PubMed, EMBASE, and Cochrane Library up to November 2021 to assess the effects of nutrients on pain, stiffness, function, quality of life, and inflammation markers. We obtained 52 references including 50 on knee OA. Twelve studies compared 724 curcumin patients and 714 controls. Using the standardized mean difference, improvement was significant with regard to pain and function in the curcumin group compared to placebo, but not with active treatment (i.e., nonsteroidal anti-inflammatory drugs, chondroitin, or paracetamol). Three studies assessed the effects of ginger on OA symptoms in 166 patients compared to 164 placebo controls. Pain was the only clinical parameter that significantly decreased. Vitamin D supplementation caused a significant decrease in pain and function. Omega-3 and vitamin E caused no changes in OA parameters. Herbal formulations effects were significant only for stiffness compared to placebo, but not with active treatment. A significant decrease in inflammatory markers was found, especially with ginger. Thus, curcumin and ginger supplementation can have a favorable impact on knee OA symptoms. Other studies are needed to better assess the effects of omega-3 and vitamin D.

## 1. Introduction

Knowledge of osteoarthritis (OA) has significantly improved in the last few decades [1,2,3]. Several treatments have been proposed for OA, including methotrexate, hydroxychloroquine, and tumor necrosis factor (TNF) inhibitors, but none are truly effective at alleviating OA symptoms [4,5,6]. Therefore, OA patients often try alternative and non-pharmacological treatments to decrease their pain and functional disability, despite little to no evidence of clinical efficacy [7,8].

Diet and nutrition have been studied in inflammatory rheumatic diseases, such as rheumatoid arthritis (RA) and spondyloarthropathies, with potential beneficial effects because of anti-inflammatory properties [9,10]. Nutrition is known to play a role not only in the emergence of rheumatic diseases, but also in their evolution and activity [11,12]. For example, omega-3 supplementation improves tender joint count, morning stiffness, and pain [13,14]. OA could now be considered an inflammatory rheumatic disease, even though it is clear that systemic inflammation in OA is less important than in RA [15,16]. Therefore, nutrition and food intake could have a positive impact on pain, stiffness, function, and other symptoms in OA patients. However, there are conflicting data concerning the effects of nutrients in OA patients. Some studies have reported beneficial effects, especially for curcumin, whereas others have reported various effects of other nutritional supplements, such as vitamin D [17,18,19].

We performed a systematic review and meta-analysis of published studies to better estimate the effects of nutrients and vitamins on symptoms, such as pain and stiffness, in patients suffering from OA.

## 2. Methods

### 2.1. Literature Search

We searched PubMed, the Cochrane Library, and the EMBASE databases to find studies that assessed the effects of nutrients on disease activity, systemic inflammation, and/or pain in OA patients over time or compared to controls until 18 November 2021. The search terms are reported in Appendix A. A manual search was also carried out by reading through the references of the analyzed studies. All studies of interest that were not found in our database search were selected for review in the next step.

### 2.2. Trial Selection

Two investigators (SM and AT) selected potentially relevant articles after reading the titles, keywords, abstracts, and full texts. Questions or doubts about article selection were resolved by consensus after discussion with the other investigator (MS). The studied population comprised patients with OA, especially knee OA, though other locations were accepted, and the analyzed intervention was oral nutrients. Routes of administration of the previous complements other than oral (e.g., transdermal, intra-articular, intravenous, and subcutaneous) were excluded. The controls were patients receiving placebo or comparator (glucosamine, chondroitin, or nonsteroidal anti-inflammatory drugs (NSAIDs)). The outcomes were symptoms of OA (total Western Ontario and McMaster Universities Arthritis Index (WOMAC)); the visual analogic scale for pain; stiffness; function; quality of life; and systemic inflammation (erythrocyte sedimentation rate (ESR) and C-reactive protein (CRP) levels). The WOMAC index, widely used in the evaluation of hip and knee osteoarthritis, is also used with other rheumatic conditions such as rheumatoid arthritis, fibromyalgia, systemic lupus erythematosus, and low back pain. It is a self-administered questionnaire consisting of 24 items divided into 3 subscales: pain, stiffness, and physical function. The sum of the scores for all three subscales gives a total WOMAC score. Higher scores on the WOMAC indicate worse pain, stiffness, and functional limitations. We also decided to collect pain scores using the most widely used scale, the visual analogic scale (VAS), a unidimensional pain intensity rating in which patients assessed their pain along a 100 mm horizontal line that ends marked with “no pain” and “worst pain ever”. All published studies concerning OA give results on pain because pain is consensual among patients reporting outcomes in OA. Only randomized controlled trials published in English or French were included in our analysis.

Data on patient characteristics were also extracted. We excluded articles without an available full-text article and, for the meta-analysis, those with data not suitable for statistical analysis (no standard deviation or interquartile range (IQR)). In the case of missing data, the corresponding author of the published article was contacted by e-mail to request the data, and the reference was ultimately excluded if we obtained no response or if the data remained missing.

### 2.3. Data Extraction

Two investigators (SM and AT) extracted the patients’ characteristics, study design information, reported outcomes, and quality assessment using a standardized data compilation form. More specifically, we extracted the number of patients treated with nutritional supplements and the number of patients treated with placebo. For these populations, we extracted the mean and standard deviation for the change in total WOMAC [20]; stiffness expressed by the WOMAC or by morning stiffness duration; pain expressed by a visual analogue scale (VAS) or Knee Injury and Osteoarthritis Outcome Score (KOOS) [21]; VAS for activity; function expressed by the WOMAC or KOOS [21]; quality of life expressed by the Medical Outcome Study Short Form 12 (SF12) [22] or KOOS [21]; and the erythrocyte sedimentation rate (ESR) and C reactive protein (CRP) levels after intervention. We also recorded age, weight, and body mass index, and the percentage of females and smokers when available.

### 2.4. Quality Assessment of the Included Studies

Risk of bias was assessed for each study during data collection using the Jadad scale [23]. Records limited to abstracts were not assessed because of the paucity of available information. We classified the studies according to Jadad score ≥4 or <4.

### 2.5. Statistical Analysis

Baseline characteristics were summarized for each study sample and reported as the mean with standard deviation or number and percentage for continuous and categorical variables, respectively. The differences between the effects of nutritional supplements and placebo or comparator with regard to pain and other parameters were expressed as standardized mean differences (SMDs) using the inverse-variance method. We chose SMD because the included studies could assess the same outcome but measure it in a variety of ways. The SMD was interpreted according to Cohen [24]: <0.2, trivial; 0.2–0.3, small; 0.5–0.8, moderate; and >0.8, large.

Statistical heterogeneity was assessed by examining forest plots, confidence intervals (CIs), and calculating the I^2^ index, which is the most common metric for measuring the magnitude of between-study heterogeneity and is easily interpretable. I^2^ values range between 0% and 100% and are typically considered low when <25%, modest when 25–50%, and high when >50%. Random effects models assuming between- and within-study variability (the DerSimonian and Laird approach) were used if heterogeneity was present; otherwise, a fixed effect model was used. When possible (i.e., sufficient sample size), meta-regression analysis was used to study the relationship between the effect sizes of nutritional supplement consumption, SMDs in VAS pain, and study characteristics (Jadad score ≥4 or <4, type, placebo or active comparator). Subgroup analyses were also performed according to the study’s characteristics.

Finally, to check the robustness of the results, sensitivity analyses were performed when the number of included studies in the meta-analysis was too low or according to funnel plots and Egger’s test. Type-I error was fixed at 5% and two-sided. All statistical analyses were performed using Stata software (version 15, StataCorp, College Station, TX, USA). All of the items required on the PRISMA checklist were fulfilled in this study. This meta-analysis was registered in PROSPERO (no. CRD42021253457).

## 3. Results

### 3.1. Literature Selection

A total of 6345 citations were obtained from the initial search (Figure 1). All randomized controlled studies exploring nutritional (curcumin, vitamin D, ginger, vitamin E, omega-3, herbal formulations, multivitamin, and others) effects were included. Of these, 57 studies were eligible for inclusion in the meta-analysis. Four other references were found from grey literature. Of the 61 studies of interest, 9 were excluded because the published data were not usable for statistical analysis. Characteristics of the 52 included studies are provided in Table 1.

The quality assessment of 51 included studies, as 1 abstract (Guo et al.) was excluded from the quality analysis, found a good median Jadad score of 4.5 (IQR: 3–5). Thirty-three studies (64.7%) had a Jadad score ≥4.
Figure 1Preferred Reporting Items for Systematic reviews and Meta-Analysis (PRISMA) diagram.
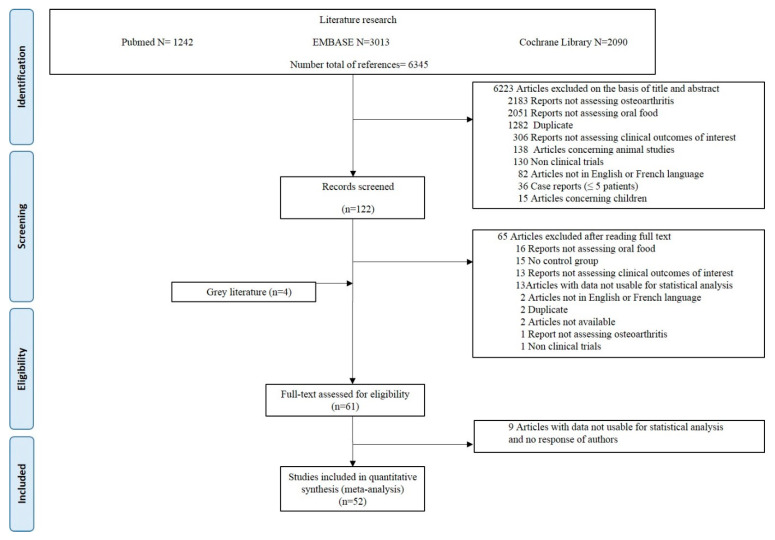


### 3.2. Characteristics of the Included Studies

The 52 studies concerned 4744 patients: 2410 in the nutrients group and 2336 in the control group (Table 1). All studies except one were published articles and concerned randomized control trials. Fifty studies included patients with knee OA and the remaining two studies concerned patients with OA without a precise articular location. The control was a placebo in 36 studies (*n* = 1574 patients total) and an active treatment in the 16 others (NSAIDs in 11 studies, and chondroitin, glucosamine, paracetamol, vitamin B, and fish oil each in 1 study). At baseline, the weighted mean age in the nutrients group (*n* = 2324) was 59.5 ± 7.8 years and 73.2% of patients (*n* = 2212) were women. Forty-two percent of patients (*n* = 974) had a Kelgren and Lawrence (KL) score of 2 and 38.8% (*n* = 1005) a KL score of 3. In the control group (*n* = 2256), the weighted mean age was 59.3 ± 7.8 years and 71.1% of patients (*n* = 2157) were women. At baseline, the weighted mean body mass index (BMI) was 27.7 ± 4.2 kg/m^2^ in the nutrients group (*n* = 1466) and 27.6 ± 4.5 kg/m^2^ in the control group (*n* = 1407). The characteristics of the patients included in the selected studies are presented in Appendix A.
nutrients-14-01607-t001_Table 1Table 1Characteristics of the publications included in the meta-analysis.StudyDiseaseNo. of ParticipantsInterventionDoseDuration (wks)ControlJadad ScoreMeasuresInterventionTotalAtabaki [25]KOA1515Curcumin80 mg/d12Placebo5VAS pain, ESR, CRPHaroyan [26]KOA6668Curcumin1500 mg/d12Placebo5Total WOMAC, VAS pain, ESR, CRPHenrotin [27]KOA4945Curcumin380 mg/d12Placebo5KOOS, function, QoLKhanna [28]KOA4040CurcuminNA12Chondroitin3Total WOMAC, stiffness, VAS painKuptniratsaikul 2014 [29]KOA171160Curcumin1500 mg/d4NSAID5Total WOMAC, WOMAC function, stiffness, VAS painKuptniratsaikul 2009 [30]KOA5255Curcumin2000 mg/d6NSAID4VAS painMadhu [31]KOA2929Curcumin1000 mg/d6Placebo4Total WOMAC, VAS painPanda [32]KOA2525Curcumin500 mg/d12Placebo5Total WOMAC, WOMAC function, stiffness, VAS painShep [33]KOA7169Curcumin500 mg/d4NSAID1VAS pain, function, QoLSinghal [34]KOA7371Curcumin1000 mg/d6Paracetamol3Total WOMACSrivastava [35]KOA7882CurcuminNA16Placebo5Stiffness, function, VAS painWang [36]KOA3634Curcumin1000 mg/d12Placebo5Stiffness, function, VAS pain, QoLPanahi [37]KOA1921Curcumin1500 mg/d6Placebo4Total WOMAC, WOMAC function, stiffness, VAS painAltman [38]KOA247124GingerNA6Placebo5Total WOMAC, WOMAC function, stiffness, VAS pain, QoLWigler [39]KOA2914Ginger250 mg/d12Placebo4VAS pain, VAS activityBolognesi [40]KOA5428GingerNA24Placebo0VAS pain, stiffness, function, ESR, CRPJin [41]KOA413209Vitamin D350,000 IU/m96Placebo5Total WOMAC, WOMAC function, stiffness, VAS painMacAlindon [42]KOA14673Vitamin D32000 IU/d96Placebo5VAS pain, functionSanghi [43]KOA10352Vitamin D360,000 IU/m48Placebo5Total WOMAC, WOMAC function, stiffness, VAS painMedhi [44]KOA5050Vitamin E200 IU/d8Placebo2VAS painDehghan [45]KOA3538Vitamin ENA3Placebo2Stiffness, function, VAS painTantavisut [46]KOA3135Vitamin E400 IU/d8Placebo5Stiffness, function, VAS painWluka [47]KOA6769Vitamin E500 IU/d96Placebo5Total WOMAC, WOMAC function, stiffness, VAS painEssouiri [48]KOA5149Vitamin ENA8Placebo1Stiffness, function, VAS painColker [49]KOA1615MultivitaminNA6Placebo4Total WOMAC, VAS pain, KOOS for pain, KOOS for function, KOOS for QoLFrestedt [50]KOA2016MultivitaminNA12Placebo5Stiffness, function, VAS painJacquet [51]KOA4140Omega-3NA12Placebo5Total WOMAC, WOMAC function, stiffness, VAS painStammers [52]OA4443Omega-3NA24Placebo3VAS painChopra [53]KOA4545HerbalNA32Placebo3Stiffness, VAS painGuo [54]KOA2121HerbalNA2NSAIDabsVAS painTao [55]KOA4545HerbalNA8Glucosamine1Total WOMAC, VAS pain, ESR, CRPWu [56]KOA3020HerbalNA2NSAID1Stiffness, VAS painFarpour [57]KOA1820HerbalNA8NSAID4Total WOMAC, WOMAC function, stiffness, VAS painGupta [58]KOA4444HerbalNA12Placebo3VAS painHamblin [59]KOA95HerbalNA10Placebo4Total WOMAC, WOMAC function, stiffness, VAS painKarimifar [60]KOA2626HerbalNA4NSAID3VAS pain, VAS activityKarlapundi [61]KOA3432HerbalNA12Placebo3Total WOMAC, WOMAC function, stiffness, VAS painKoonrungsesomboon [62]KOA100100HerbalNA4NSAID5Stiffness, VAS pain, KOOS for pain, KOOS for function, KOOS for QoLMoré [63]KOA4644HerbalNA12Placebo3Total WOMAC, stiffness, VAS painThomford [64]OA4411HerbalNA8NSAID4VAS painKakatum [65]KOA3231HerbalNA4NSAID5Total WOMAC, WOMAC function, stiffness, VAS painPinsornsak [66]KOA3130HerbalNA4NSAID5Total WOMAC, WOMAC function, stiffness, VAS painLiu [67]HOA5254HerbalNA12Placebo5VAS pain, VAS activity, function, QoLSchumacher [68]KOA2622CherryNA6Placebo5StiffnessPuente [69]KOA3030BeeswaxNA6Placebo4Total WOMAC, WOMAC function, stiffness, VAS painShin [70]KOA2624Deer boneNA12Placebo5Total WOMAC, WOMAC function, stiffness, VAS painSalimzadeh [71]KOA3937GarlicNA12Placebo5Total WOMAC, WOMAC function, stiffness, VAS painLau [72]KOA4040Green musselNA24Placebo3VAS pain, VAS activity Zawadzki [73]KOA2525Green musselNA12Fish oil5VAS pain, VAS activity May [74]KOA3837MelonNA12Placebo5KOOS for pain, KOOS for function, KOOS for QoLSadat [75]KOA2223SesameNA8Placebo1VAS painSchell [76]KOA89StrawberriesNA12Placebo5VAS pain, CRPKOA = knee osteoarthritis; HOA = hand osteoarthritis; VAS = visual analog scale; CRP = C reactive protein; ESR = erythrocyte sedimentation rate; NA = not available; QoL = quality of life; NSAID = nonsteroidal anti-inflammatory drug; abs = abstract; d = day.

### 3.3. Overall Effects of Nutritional Supplements on OA Symptoms

#### 3.3.1. Vitamin D

Three studies assessed the effects of vitamin D on OA symptoms. These studies comprised 334 vitamin D patients and 328 placebo controls. The dose of vitamin D received by 506 patients varied from 2000 to 3000 IU per day for 1 or 2 years. VAS pain and WOMAC function were significantly decreased, with small improvement. No changes were found in total WOMAC or stiffness (Appendix A). The ESR and CRP levels were not analyzed because of an insufficient number of patients for the meta-analysis.

#### 3.3.2. Curcumin

Twelve studies assessed the effects of curcumin supplementation on OA symptoms, comprising 705 curcumin patients and 693 controls (298 placebo and 395 active control). The weighted mean dose of curcumin received by 506 patients was 1000 mg per day for 4 to 16 weeks. All clinical symptoms of OA, except quality of life, improved with supplementation when pooling all studies and timepoints. Improvement was significantly large for VAS pain and WOMAC function, moderate for total WOMAC, and nearly significant for WOMAC stiffness (Table 2 and Figure 2 and Figure 3). Other forest plots are in supplementary files (Appendix A). We found a conflicting effect of curcumin supplementation on systemic inflammation markers, with a significantly large decrease in the ESR but not in the CRP level (Table 2). Egger’s test was significant for total WOMAC (*p* = 0.02). Sensitivity analysis found a consistently significant decrease in total WOMAC, but the improvement was smaller (SMD = −0.34 (95% CI −0.67; −0.01)) according to the funnel plot when the studies by Madhu [31] and Khanna [28] were excluded (Appendix A). Other sensitivity analyses found no change in the results of curcumin on VAS pain and WOMAC function.
nutrients-14-01607-t002_Table 2Table 2Effects of nutritional supplements on OA parameters over time.MeasureTime PointVitamin DCurcuminGingerVitamin EHerbalMultivitaminOmega-3Others**WOMAC total**Overall1 month3 months6 months12 months24 months−0.92 (−2.32; 0.48)I^2^ = 97%**−1.65 (−2.10; −1.20)****−0.22 (−0.42**; −0.03)**−0.78 (−1.31; −0.24)**I^2^ = 92%**−0.77 (−1.53; −0.01)**−0.80 (−1.84; 0.24)−0.19 (−0.44; 0.06)I^2^ = NA−0.19 (−0.44; 0.06)0.26 (−0.08; 0.60)I^2^ = NA0.26 (−0.08; 0.60)−0.62 (−1.26; 0.02)I^2^ = 88%−0.08 (−0.43; 0.27)−1.02 (−2.10; 0.07)2.19 (1.29; 3.09)I^2^ = NA2.19 (1.29; 3.09)−1.24 (−1.71; −0.76)I^2^ = NA−1.24 (−1.71; −0.76)**−1.24 (−1.85; −0.63)**I^2^ = 78%**−1.64 (−2.12; −1.17)****−0.85 (−1.57; −0.13)****WOMAC stiffness**Overall1 month3 months6 months12 months24 months−0.07 (−0.25; 0.10)I^2^ = 0%0.05 (−0.33; 0.44)−0.11 (−0.30; 0.09)−0.61 (−1.29; 0.06)I^2^ = 94%−0.26 (−0.89; 0.38)**−0.95 (−1.87; −0.03)**−2.50 (−5.13; 0.13)I^2^ = 98%−1.77 (−4.75; 1.20)**−3.97 (−4.90; −3.04)**−1.04 (−2.31; 0.24)I^2^ = 97%0.13 (−0.33; 0.59)−2.38 (−6.50; 1.74)0.19 (−0.15; 0.53)**−0.71 (−1.23; −0.19)**I^2^ = 89%−0.03 (−0.29; 0.23)**−1.28 (−1.95; −0.62)****−1.81 (−2.40; −1.21)**−0.63 (−1.47; 0.22)I^2^ = NA−0.63 (−1.47; 0.22)−0.94 (−1.40; −0.48)I^2^ = NA−0.94 (−1.40; −0.48)**−0.90 (−1.37; −0.44)**I^2^ = 71%**−0.91 (−1.49; −0.33)**−0.92 (−1.98; 0.14)**VAS pain**Overall1 month3 months6 months12 months24 months**−0.20 (−0.35; −0.04)**I^2^ = 0%−0.13 (−0.51; 0.26)**−0.21 (0.38; −0.04)****−1.81 (−2.93; −0.69)**I^2^ = 98%−0.10 (−1.34; 1.15)**−4.5 (−6.85; −2.15)****−3.76 (−6.88; −0.65)**I^2^ = 98%−3.64 (−10.47; 3.19)−0.28 (−1.09; 0.53)**−7.79 (−9.38; −6.20)**−0.84 (−1.75; 0.05)I^2^ = 96%0.23 (−0.07; 0.53)**−2.06 (−3.91; −0.20)**0.22 (−0.11; 0.56)−0.28 (−0.76; 0.19)I^2^ = 88%−0.02 (−0.61; 0.57)−0.04 (−0.42; 0.35)**−2.78 (−3.49; −2.08)****−1.38 (−2.16; −0.59)**I^2^ = NA**−1.38 (−2.16; −0.59)**−0.51 (−1.92; 0.90)I^2^ = 94%**−1.23 (−1.71; −0.75)**0.21 (−0.31; 0.73)**−1.30 (−1.93; −0.66)**I^2^ = 92%**−1.24 (−2.44; −0.05)****−1.46 (−2.50; −0.41)****−0.81 (−1.27; −0.35)****KOOS pain**Overall1 month3 months6 months12 months24 months




1.19 (0.42; 1.96)I^2^ = NA1.19 (0.42; 1.96)
0.33 (−0.13; 0.78)I^2^ = NA0.33 (−0.13; 0.78)**VAS activity**Overall1 month3 months6 months12 months24 months

−0.31 (−1.12; 0.50)I^2^ = NA−0.31 (−1.12; 0.50)
−0.02 (−0.40; 0.36)I^2^ = NA−0.02 (−0.40; 0.36)

**−1.23 (−2.20; −0.26)**I^2^ = 94%−1.24 (−4.01; 1.54)−1.61 (−3.68; 0.46)**−0.61 (−1.06; −0.16)****WOMAC function**Overall1 month3 months6 months12 months24 months**−0.44 (−0.80; −0.09)**I^2^ = 76%**−0.90 (−1.31; −0.50)****−0.44 (−0.80; −0.09)****−1.43 (−2.59; −0.27)**I^2^ = 98%−0.47 (−1.18; 0.25)−2.76 (−6.31; 0.79)−11.8(−23.57; 0.02)I^2^ = 99%−12.1(−35.87; 11.61)**−11.5(−13.75; −9.21)**−0.92 (−2.33; 0.49)I^2^ = 97%0.75 (0.27; 1.22)−2.41 (−5.36; 0.54)0.25 (−0.08; 0.59)−0.47 (−1.07; 0.12)I^2^ = 87%−0.11 (−0.42; 0.19)−0.71 (−1.88; 0.46)−0.32 (−1.15; 0.51)I^2^ = NA−0.32 (−1.15; 0.51)**−1.21 (−1.69; −0.74)**I^2^ = NA**−1.21 (−1.69; −0.74)**−0.92 (−1.89; 0.04)I^2^ = 89%**−1.83 (−2.43; −1.22)**−0.47 (−1.13; 0.19)**KOOS function**Overall1 month3 months6 months12 months24 months




**2.97 (1.93; 4.01)**I^2^ = NA**2.97 (1.93; 4.01)**
**0.52 (0.20; 0.85)**I^2^ = 0%0.41 (−0.05; 0.86)**0.64 (0.18; 1.10)****CRP**Overall1 month3 months6 months12 months24 months
−1.32 (−3.23; 0.58)I^2^ = 93%−1.32 (−3.23; 0.58)**−1.36 (−1.80; −0.92)**I^2^ = 38%**−1.84 (−2.48; −1.20)****−1.12 (−1.69; −0.54)****−1.19 (−1.77; −0.61)**
−0.19 (−0.60; 0.23)I^2^ = NA−0.19 (−0.60; 0.23)

−0.48 (−1.45; 0.48)I^2^ = NA−0.48 (−1.45; 0.48) **ESR**Overall1 month3 months6 months12 months24 months
**−0.92 (−1.73; −0.10)**I^2^ = 75%**−0.92 (−1.73; −0.10)****−2.13 (−3.37; −0.89)**I^2^ = 90%**−2.63 (−3.37; −1.90)****−0.98 (−1.55; −0.42)****−2.84 (−3.60; −2.08)**
−0.07 (−0.48; 0.35)I^2^ = NA−0.07 (−0.48; 0.35)


**SF12**Overall1 month3 months6 months12 months24 months
0.54 (−0.25; 1.32)I^2^ = 91%1.27 (0.91; 1.64)0.17 (−0.14; 0.49)0.08 (−0.17; 0.33)I^2^ = NA0.08 (−0.17; 0.33)
−0.01 (−0.39; 0.37)I^2^ = NA−0.01 (−0.39; 0.37)


**KOOS QoL**Overall1 month3 months6 months12 months24 months




0.61 (−0.11; 1.33)I^2^ = NA0.61 (−0.11; 1.33)
**0.64 (0.26; 1.03)**I^2^ = 26%0.45 (−0.01; 0.91)**0.84 (0.37; 1.32)**Values are given as standardized mean differences (confidence intervals). NA = not available; QoL = quality of life. Significant data are in bold.
Figure 2Forest plot of the effects of curcumin on pain. ID = study identity; SMD = standard mean difference; CI = confidence interval.
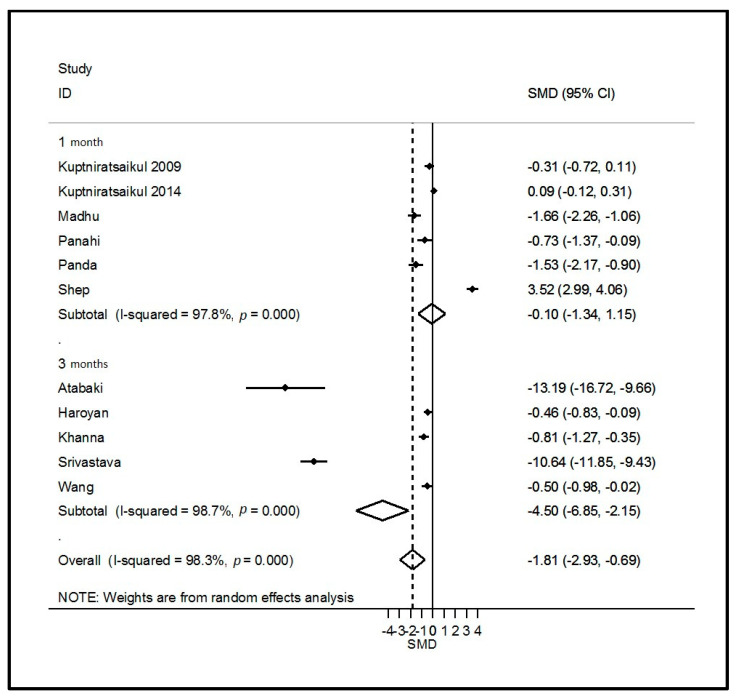

Figure 3Overall effects of nutritional supplements in osteoarthritis patients compared to controls. Data are shown as the standardized mean differences (SMDs) and confidence intervals. VAS = visual analogic scale; CRP = C reactive protein; ESR = erythrocyte sedimentation rate; WOMAC = Western Ontario and McMaster Universities Osteoarthritis Index.
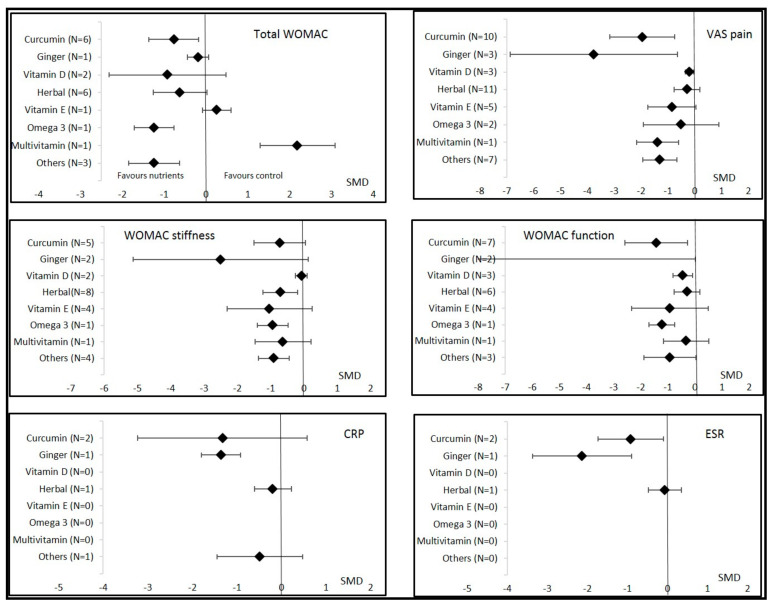


#### 3.3.3. Ginger

Three studies assessed the effects of ginger on OA symptoms, comprising 166 ginger patients and 164 placebo controls. The dose was specified in only one study and was 250 mg per day for a duration of 12 weeks. Other supplementation durations were 6 and 24 weeks. VAS pain was the only clinical parameter that significantly decreased with ginger intake (SMD = −3.76 (95% CI −6.88; −0.65); Table 2 and Figure 2 and Figure 3). Sensitivity analysis according to funnel plots found different results for the VAS pain analysis after excluding the study by Bolognesi [40] (SMD = −0.20 (95% CI −0.44; 0.04)) (Appendix A). However, Egger’s test was not significant for the VAS pain analysis (*p* = 0.13) with the results from the study by Bolognesi [40]. We also found a favorable effect of ginger on systemic inflammation, with a significantly large decrease in the ESR and CRP levels (Table 2 and Figure 3). Egger’s test was significant for the ESR and CRP analysis (*p* = 0.01) with the results of the study by Bolognesi [40] (Appendix A). No sensitivity analyses could be performed because it was the only study included at a different time of assessment.

#### 3.3.4. Vitamin E and Multivitamin

The effects of vitamin E supplementation were assessed in five studies, comprising 234 vitamin E patients and 241 controls (203 placebo and 38 active controls). The dose of vitamin E was 400 or 500 IU per day during 3, 8, or 96 weeks. We found no significant improvement in all clinical OA parameters or the ESR and CRP levels (Table 2) (Appendix A). Two other studies assessed the effects of multivitamin supplementation in 36 patients compared to 31 placebo controls. VAS pain was significantly decreased and the KOOS for function significantly improved.

#### 3.3.5. Herbal Formulations

Fifteen studies assessed the effects of different herbal formulations consisting of traditional officinal plants from ayurvedic medicine (e.g., Boswellia, Curcuma, Urtica dioica, Piper Nigrum, and Tinospora cordifolia) providing a large board of bioactive phytochemicals on OA symptoms. These studies comprised 577 herbal patients and 528 controls (224 placebo and 304 active controls). These studies were carried out mostly in China, Thailand, India, and Iran. Nine of the studies lasted 2 to 8 weeks, five 10–12 weeks (*n* = 5), and one 32 weeks. WOMAC stiffness was the only clinical parameter that significantly decreased, with moderate improvement (Table 2) (Appendix A). The safety of herbal formulations was good. No severe drug side effects were observed in the included studies. Some gastrointestinal side effects, such as nausea, abdominal pain, stomach burning, dyspepsia, diarrhea, and loss of appetite, occurred sporadically and at the same frequency in the control group.

#### 3.3.6. Omega-3

Only two studies assessed the effects of omega-3 intake on OA parameters, comprising 85 omega-3 patients and 83 placebo controls. The dose of omega-3 per day was not available and supplementation lasted 12 or 24 weeks. WOMAC function was significantly improved but other parameters, especially VAS pain, were not significantly modified.

#### 3.3.7. Other Nutritional Supplementations

Nine studies reported effects of other nutritional strategies including strawberries, deer bone, melon, sesame, cherry, garlic, and green mussel. They found a significant improvement in total WOMAC, stiffness, VAS pain, and quality of life. The KOOS for function, but not WOMAC function, significantly improved. No change in CRP levels was noted.

### 3.4. Effects of Nutritional Supplements Depending on Type of Controls and Jadad Score

Sub-group analyses were possible for herbal formulations, curcumin, and vitamin E. For other dietary supplements (vitamin D, omega-3, ginger, and multivitamin), the data were insufficient to perform this analysis. We found that the effects of herbal formulations and curcumin on OA parameters were not significant or relevant if the control was an active treatment (NSAID, chondroitin, glucosamine, or paracetamol) (Figure 4). The effects of herbal formulations were no more significant with regard to WOMAC stiffness if the quality of the studies was higher (Jadad score ≥4). Similarly, WOMAC function significantly improved with herbal formulations if the Jadad score was <3 (SMD = −2.04 (95% CI −2.55; −1.53)) but not if it was >4 (SMD = −0.08 (95% CI −0.30; 0.145)). The effects of curcumin were significant with regard to total WOMAC, VAS pain, and WOMAC function if the Jadad score was >4, but no more significant on these parameters if the Jadad score was <3 (Figure 5). Changes in the Jadad score or type of controls did not modify the results for the effects of vitamin E on OA parameters.

## 4. Discussion

This meta-analysis included 51 randomized controlled trials assessing the effects of oral nutritional supplements on the symptoms of OA, particularly knee OA. The analysis showed that curcumin supplementation had a beneficial effect on pain, total WOMAC, and WOMAC function. Curcumin is an abundant polyphenol in the ground rhizome of Curcuma longa and has been used extensively in ayurvedic medicine for centuries [77]. This polyphenol exhibits in vivo pleiotropic biological activities, including antiproliferative, antioxidant, or anti-inflammatory properties that support its interest in the prevention and the treatment of OA [78]. The positive results with curcumin supplementation were recently confirmed by Hsiao et al., who concluded that a beneficial effect of curcumin is present, especially at a high dose (i.e., ≥1000 mg per day) [79]. Zheng et al. added that this beneficial effect needed a curcumin supplementation for more than 12 weeks [80]. This interesting effect on VAS pain and WOMAC function seemed to be mediated by anti-inflammatory properties, with a significant decrease in the ESR. Curcuma, a rooted plant in the ginger family, has been recognized to have anti-inflammatory and antioxidant properties. One main effect is the inhibition of the production of cytosolic phospholipase A2 (CPLA2) or cyclooxygenase 2 (COX2), providing active nitrogen and inhibition of oxidation by removing free radicals [81,82,83,84].

Curcumin was significantly effective compared to placebo, but not if the control was NSAIDs. Therefore, the nature of the controls had an impact on the interpretation of the curcumin results. No difference was found in pain, stiffness, or function when the comparator was active treatment, whereas differences were significantly in favor of curcumin in the case of placebo. Analysis of the comparator (active or placebo) is important to interpret the effects of dietary supplements in OA patients. The methodological quality of the included studies also had an important impact the interpretation of the results observed after supplementation with curcumin. We found that studies with lower methodological quality (Jadad score <3) tended to favor the comparator, whereas studies with a Jadad score >4 showed results in favor of curcumin.

Ginger intake also caused significant improvement in the ESR and CRP levels, which was associated with the greatest improvement in pain (SMD = −3.76). Sensitivity analyses slightly moderated these very favorable results with a weak decrease in pain when the study by Bolognesi [40] was removed. However, several studies confirmed that ginger can reduce inflammatory markers (CRP and concentration of nitric oxide) or proinflammatory cytokines [85,86,87]. Ginger (Zingiber officinale Rosc.) is a rhizome that is rich in phytochemicals, including gingerol, endowed with health-benefiting properties of interest to manage diseases related to inflammation and oxidative stress, including OA [88]. Ginger intake can reduce pain due to its reported antioxidant activity, the inhibition of transcription factor Nf-kB, or by an agonist action on the vanilloid nociceptor [89]. However the mechanism of this benefic effect of ginger is still much debated and still quite unclear. Naderi et al. concluded that ginger can be recommended as suitable supplementation for OA patients at a dose of 1 gram per day [85]. This significant improvement in pain was also reported for oral ginger by Araya-Quintanilla et al. in 2020, with good tolerance [90]. However, they only included two studies in their meta-analysis and concluded that other evidence is needed to support the use of oral ginger.

In addition to its primary functions in calcium homeostasis and in the regulation of bone metabolism, an anti-inflammatory and immunomodulatory role has also been reported for vitamin D [91,92,93]. However, in RA and spondyloarthritis, oral vitamin D supplementation has shown a limited effect [94,95]. In the present meta-analysis, we did not find any significant improvement with vitamin D supplementation with regard to total WOMAC or stiffness. Improvement in pain and WOMAC function was weak. This can be explained by the low number of studies (*n* = 3). However, the analysis is in accordance with the review published by Hussain et al. in 2017, which concluded that there is insufficient evidence to support the use of vitamin D supplementation for patients with knee OA [96].

Vitamin E refers to tocopherols and tocotrienols, as other vitamins are strong antioxidant found in plants. A low circulating level of vitamin E was observed in human OA patients compared to healthy volunteers [97]. In preclinical studies, supplementation with vitamin E was associated with a decrease in oxidative stress [98,99]. However, evidence did not support the use of vitamin E in clinical practice [100,101] except in association with other nutrients [102]. The effect of vitamin E was also disappointing in the current meta-analysis, with no significant improvement in pain. Omega-3 supplementation was also very promising for OA because it was recently added to the management guidelines for RA [9]. Supplementation by capsule, favoring fish oils (mainly EPA and DHA) with dosages of 2–3 g per day for 3 to 6 months, could be proposed for symptomatic relief in patients who have RA.

Similar to the results for vitamin E, the results for omega-3 (fish oil) supplementation were inconclusive, with no significant improvement of OA symptoms in our meta-analysis. However, only two studies in OA patients were included in this meta-analysis, whereas the literature on the effects of omega-3 in RA patients is more extensive, with more than 40 studies. Other studies will be needed to draw more relevant conclusions about the omega-3 supplementation in OA patients.

Drawing strong conclusions on the effects of herbal formulations was very difficult. We found that herbal formulations significantly improved stiffness, but not pain, total WOMAC, or function. The first major factor in interpreting this result is the controls used in randomized controlled trials. Studies on herbal formulations were considered negative when the comparator was an active compound, such as NSAIDs or glucosamine. When the comparator was a placebo control, supplementation with an herbal formulation was associated with a beneficial effect on total WOMAC and pain. Thus, the effect of herbal formulations is greater than placebo but not different from that of NSAIDs or glucosamine. Moreover, the safety of the herbal formulations was very good, with no severe side effects and only sporadic and transient gastrointestinal effects. Therefore, herbal formulations have a positive risk–benefit balance and could be proposed to OA patients to alleviate their clinical symptoms. However, the second important limitation of these interventional studies is the varied composition of the herbal formulation. Indeed, all these products originate from ayurvedic medicine are rarely used in other countries than China, Thailand, or India. Although the mechanism of action of herbal medicine is poorly understood, even if we can suppose that its supported by the presence of phytochemicals (e.g. polyphenols, alkaloids, and terpenes), this alternative treatment seems to be increasingly considered in the management of pain [103]. As mentioned above, the methodological quality of the included studies had an important impact on the interpretation of the results. We found that studies with a Jadad score <3 were more willing to show a significant improvement in OA symptoms for herbal formulations though higher quality studies found no difference from the comparator.

Our meta-analysis has limitations. First, the number of OA patients in the included studies is low, especially for omega-3, vitamin D, and multivitamin supplementation. Sensitivity analyses showed a change in the effects of ginger on VAS pain and concluded that the robustness of some results was rather low. However, several results did not change during the sensitivity analyses, especially those for curcumin with regard to VAS pain or WOMAC function, which suggests that these results are still reliable and can be considered relevant. The heterogeneity between the included studies was mostly high with an I^2^ > 80%. This was the case for studies investigating the effects of sesame, strawberries, and garlic. Another limitation is related to publication bias, as positive studies are more likely to be published than negative studies. We cannot exclude that some investigations found no significant effects of nutritional supplements on OA symptoms but were never published. In the same way, only one study each concerned garlic, cherry, or deer bone supplementation, with a significant positive effect, but other studies are necessary to confirm these results.

Plants from ayurvedic medicine and from our food are rich sources of a large board of bioactive compounds including alkaloids, terpenes, polyphenols, fatty acids, etc., which possess interesting properties in terms of the management of OA. For now, curcumin or ginger supplementation appears to be of the most interest in terms of improving knee OA symptoms. The effects of other nutrients or nutritional supplements (omega 3, vitamins E and D) are disappointing. However, the number of relevant studies is insufficient. Further studies, such as randomized controlled trials, to assess the effects of nutrients compared with placebo or active comparator (NSAID, chondroitin, glucosamine, or paracetamol) could permit the drawing of relevant conclusions on the place of nutritional supplementation in osteoarthritis management.

## Figures and Tables

**Figure 4 nutrients-14-01607-f004:**
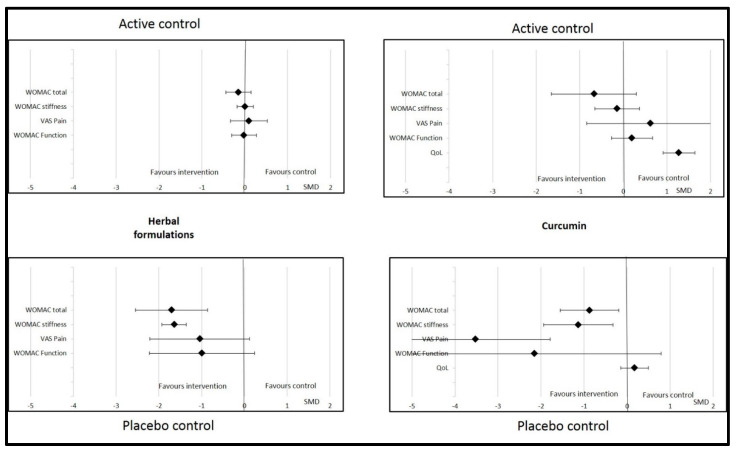
Effects of curcumin and herbal formulations on osteoarthritis parameters depending on type of control. Data are presented at standardized mean differences (SMDs) and confidence intervals. VAS = visual analogic scale; WOMAC = Western Ontario and McMaster Universities Osteoarthritis Index; QoL = quality of life.

**Figure 5 nutrients-14-01607-f005:**
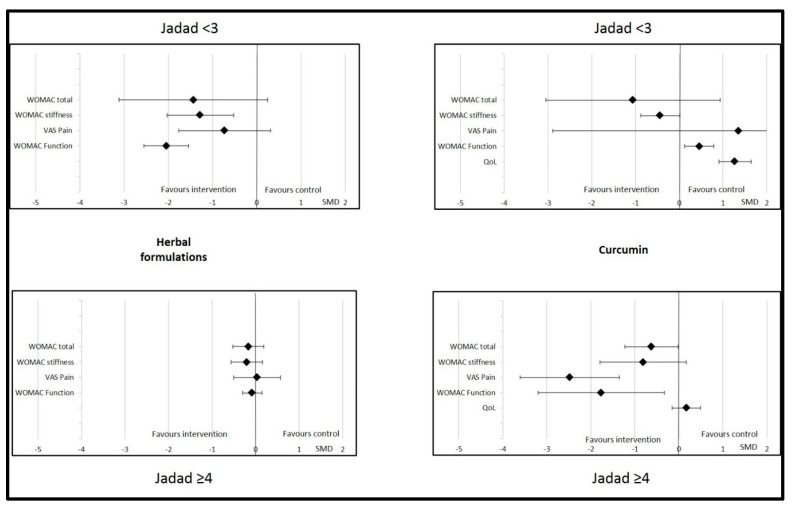
Effects of curcumin and herbal formulations on osteoarthritis parameters depending on Jadad score. Data are presented at standardized mean differences (SMDs) and confidence intervals. VAS = visual analogic scale; WOMAC = Western Ontario and McMaster Universities Osteoarthritis Index. QoL = quality of life.

## Data Availability

Data are available on reasonable request to Mathieu.

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
