# Peer review of "A Meta-Analysis of the Impact of Nutritional Supplementation on Osteoarthritis Symptoms"

_nutrients, 2022, doi:10.3390/nu14081607_

Round 1

Reviewer 1 Report

Comments to the Authors of manuscript number: nutrients-1641272 entitled “A meta-analysis of the impact of nutritional supplementation on osteoarthritis symptoms.”.

The authors have presented a meta-analysis focused on the nutritional supplements and OA. Authors have presented in clear manner the choose of studies, which were further subjected to statistical analysis, which involved the results of many scientific nutritional studies concerning many symptoms of OA (pain, stiffens, function, quality of life, basal parameters of systemic inflammation). Meta-analysis is performed when all studies have addressed to the same problem. Chosen studies had some degree of error, and these condition allow Authors to perform presented meta-analysis.

Authors in good manner contrasted results from various studies and identified patterns among all these study results.

  1. The manuscript appears well written and well structured.
  2. The abstract concisely provides the overview.
  3. L 70-71 – please explain why these two parameters were chosen
  4. The statistical analysis is very well in details described.
  5. L 69 – as it is a journal not only for doctors, I propose to explain in more detail what WOMAC and VAS are.

e.g. That WOMAC Index is developed for hip and knee osteoarthritis, and it is also used with other rheumatic conditions such as: rheumatoid arthritis, juvenile rheumatoid arthritis, fibromyalgia, systemic lupus erythematosus and low back pain.

  1. very good meta-analysis.

Author Response

Reviewer 1

Comments to the Authors of manuscript number: nutrients-1641272 entitled “A meta-analysis of the impact of nutritional supplementation on osteoarthritis symptoms.”.

The authors have presented a meta-analysis focused on the nutritional supplements and OA. Authors have presented in clear manner the choose of studies, which were further subjected to statistical analysis, which involved the results of many scientific nutritional studies concerning many symptoms of OA (pain, stiffens, function, quality of life, basal parameters of systemic inflammation). Meta-analysis is performed when all studies have addressed to the same problem. Chosen studies had some degree of error, and these condition allow Authors to perform presented meta-analysis.

Authors in good manner contrasted results from various studies and identified patterns among all these study results.

  1. The manuscript appears well written and well structured.
  2. The abstract concisely provides the overview.
  3. L 70-71 – please explain why these two parameters were chosen
  4. The statistical analysis is very well in details described.
  5. L 69 – as it is a journal not only for doctors, I propose to explain in more detail what WOMAC and VAS are.

e.g. That WOMAC Index is developed for hip and knee osteoarthritis, and it is also used with other rheumatic conditions such as: rheumatoid arthritis, juvenile rheumatoid arthritis, fibromyalgia, systemic lupus erythematosus and low back pain.

  1. very good meta-analysis.

Answer: We thank you for the positive comments (N°1, 2, 4 and 6). We decided to choose C reactive protein (CRP) and erythrocyte sedimentation rate (ESR) to assess the degree of systemic inflammation. These two parameters are relevant because they are consensual for the entire medical community.

We agree that WOMAC and VAS could be more explained for all the readers. The

Reviewer 1

Comments to the Authors of manuscript number: nutrients-1641272 entitled “A meta-analysis of the impact of nutritional supplementation on osteoarthritis symptoms.”.

The authors have presented a meta-analysis focused on the nutritional supplements and OA. Authors have presented in clear manner the choose of studies, which were further subjected to statistical analysis, which involved the results of many scientific nutritional studies concerning many symptoms of OA (pain, stiffens, function, quality of life, basal parameters of systemic inflammation). Meta-analysis is performed when all studies have addressed to the same problem. Chosen studies had some degree of error, and these condition allow Authors to perform presented meta-analysis.

Authors in good manner contrasted results from various studies and identified patterns among all these study results.

  1. The manuscript appears well written and well structured.
  2. The abstract concisely provides the overview.
  3. L 70-71 – please explain why these two parameters were chosen
  4. The statistical analysis is very well in details described.
  5. L 69 – as it is a journal not only for doctors, I propose to explain in more detail what WOMAC and VAS are.

e.g. That WOMAC Index is developed for hip and knee osteoarthritis, and it is also used with other rheumatic conditions such as: rheumatoid arthritis, juvenile rheumatoid arthritis, fibromyalgia, systemic lupus erythematosus and low back pain.

  1. very good meta-analysis.

Answer: We thank you for the positive comments (N°1, 2, 4 and 6). We decided to choose C reactive protein (CRP) and erythrocyte sedimentation rate (ESR) to assess the degree of systemic inflammation. These two parameters are relevant because they are consensual for the entire medical community.

We agree that WOMAC and VAS could be more explained for all the readers. The proposition for WOMAC index explanation has been included as required in the revised manuscript. We also clarified the definition of VAS pain. All published studies regarding OA focus on pain evaluation because it is a consensual outcome reported by OA patients. The scale mostly used to evaluate pain in OA study is the Visual Analogic Scale (VAS) that justifies its report in the present meta-analysis.

Reviewer 2 Report

Comments to the Authors: Manuscript ID: nutrients-1641272
Title: A meta-analysis of the impact of nutritional supplementation on osteoarthritis symptoms.

The authors presented the thorough review on the effects of nutrients intake in osteoarthritis patients. They performed a systematic review and meta-analysis of published studies and discussed the effects of different nutrients on pain, stiffness, function, quality of life, and inflammation markers.

The topic is interesting and actual. The presented research is comprehensive and precise. The review presents the current knowledge in a quite easy way, available for the large group of readers.

Author Response

Reviewer 2

Comments to the Authors: Manuscript ID: nutrients-1641272

Title: A meta-analysis of the impact of nutritional supplementation on osteoarthritis symptoms.

The authors presented the thorough review on the effects of nutrients intake in osteoarthritis patients. They performed a systematic review and meta-analysis of published studies and discussed the effects of different nutrients on pain, stiffness, function, quality of life, and inflammation markers.

The topic is interesting and actual. The presented research is comprehensive and precise. The review presents the current knowledge in a quite easy way, available for the large group of readers.

Answer:  We are grateful to the reviewer for the positive comments.

Reviewer 3 Report

This manuscript titled “A meta-analysis of the impact of nutritional supplementation on osteoarthritis symptoms”. The comments for this manuscript are as follows:

First of all, the authors did not deeply discuss the role of each nutritional supplement in the writing of this manuscript, and the empty content is not helpful to the reader.

The second is that what is the author's viewpoint to select these nutritional supplements as the target. From the perspective of anti-inflammatory, is vitamin C not enough? There are also many different herbal formulas around the world. Supplies for review?

The conclusions are even more lackluster, coupled with inconsistent formatting in the references.

The format of the references in this manuscript is very messy. The every first word in the titles of the cited references should not be capitalized. The format of the references should be written in accordance with the journal's regulations. There are too many errors in the section of references, please correct them.

Author Response

Reviewer 3

This manuscript titled “A meta-analysis of the impact of nutritional supplementation on osteoarthritis symptoms”. The comments for this manuscript are as follows:

First of all, the authors did not deeply discuss the role of each nutritional supplement in the writing of this manuscript, and the empty content is not helpful to the reader.

Answer:  In our manuscript, we chose to expose each nutritional supplement separately, either by compounds or by compound family (curcumin, ginger, vitamin E and omega 3), to properly describe their main effects on OA. Based on the reviewer’s comments, and in order to help reader, we have included additional informations related to the nutritional supplements used in the several studies gathered in the meta-analysis. We hope that the provided information in the revised manuscript will be helpful to better understand the interest to investigate each compound in the management of OA.  

The second is that what is the author's viewpoint to select these nutritional supplements as the target. From the perspective of anti-inflammatory, is vitamin C not enough?

Answer: We completely agree that evaluating the effects of vitamin C would have been very interesting but to our knowledge and after our literature research, no study has compared the effects of vitamin C to a validated comparator in osteoarthritis. Moreover, a recent study described the composition of an anti-inflammatory diet in knee osteoarthritis containing minimal processed foods and higher amounts of “good” fats and wholefoods. The wholefoods propositions in moderate amounts were: lean meats, eggs and dairy; and those encouraged in higher amounts were: fish, fruit, vegetables, nuts and seeds. “Good fats” included monounsaturated fats with a favourable omega-6:omega-3 ratio such as fish, seeds and olive oil (Cooper BMC Musculoskeletal Dis 2022). Therefore, studying only vitamin C would perhaps be insufficient in view of this literature. Finally, others studies maybe exist on others compounds but they are not among those we wanted to study in this meta-analysis that concerned only controlled studies.

There are also many different herbal formulas around the world. Supplies for review?

Answer: We acknowledge indeed that herbal formulations are of many and different kinds around the world. This is a limitation of our study, and was discussed (L348 and 349 in the discussion section, paragraph on herbal formulations) because it increases the heterogeneity of the included studies. However, to our opinion, it is still relevant to report the effects of herbal formulations because they are widely used in various world areas with a positive benefit-risk ratio (L347 in the discussion section, paragraph on herbal formulations). We decided to replace “herbal decoction” by “herbal formulation” in the revised manuscript because it was not only decoctions in the included studies.

The conclusions are even more lackluster, coupled with inconsistent formatting in the references.

Answer: The conclusion has been changed in the revised manuscript in order to make it more interesting and relevant.

The format of the references in this manuscript is very messy. The every first word in the titles of the cited references should not be capitalized. The format of the references should be written in accordance with the journal's regulations. There are too many errors in the section of references, please correct them.

Answer:  Inconsistent formatting in the references has been corrected and the new format has been written in accordance with the journal’s regulations. Moreover, we checked again all the references, which allowed to correct a mistake. The reference Farpour et al (N°57) was wrong and was rectified.

Round 2

Reviewer 3 Report

In the new revised manuscript, I have seen the authors have made sincere replies and corrections.